# Methods of Pre-Identification of TITO Systems

Milan Saga [1], Karel Perutka [2], Ivan Kuric [1], Ivan Zajačko [1,*], Vladimír Bulej [1], Vladimír Tlach [1] and Martin Bezák [3]

1   Faculty of Mechanical Engineering, University of Zilina, Univerzitna 8215/1, 01026 Zilina, Slovakia; milan.saga@fstroj.uniza.sk (M.S.); ivan.kuric@fstroj.uniza.sk (I.K.); vladimir.bulej@fstroj.uniza.sk (V.B.); vladimir.tlach@fstroj.uniza.sk (V.T.)
2   Faculty of Applied Informatics, Tomas Bata University in Zlin, Nad Stráněmi 4511, 760 05 Zlin, Czech Republic; kperutka@utb.cz
3   VIPO, a. s., Gen. Svobodu 1069/4, 95801 Partizanske, Slovakia; martin.bezak@vipo.sk
*   Correspondence: ivan.zajacko@fstroj.uniza.sk; Tel.: +421-910956861

**Abstract:** The content of this article is the presentation of methods used to identify systems before actual control, namely decentralized control of systems with Two Inputs, Two Outputs (TITO) and with two interactions. First, theoretical assumptions and reasons for using these methods are given. Subsequently, two methods for systems identification are described. At the end of this article, these specific methods are presented as the pre-identification of the chosen example. The Introduction part of the paper deals with the description of decentralized control, adaptive control, decentralized control in robotics and problem formulation (fixing the identification time at the existing decentralized self-tuning controller at the beginning of control and at the beginning of any set-point change) with the goal of a new method of identification. The Materials and methods section describes the used decentralized control method, recursive identification using approximation polynomials and least-squares with directional forgetting, recursive instrumental variable, self-tuning controller and suboptimal quadratic tracking controller, so all methods described in the section are those ones that already exist. Another section, named Assumptions, newly formulates the necessary background information, such as decentralized controllability and the system model, for the new identification method formulated in Pre-identification section. This section is followed by a section showing the results obtained by simulations and in real-time on a Coupled Drives model in the laboratory.

**Keywords:** pre-identification; least squares method; instrumental variable method; robotics; sensor

## 1. Introduction

Most processes in practice are processes that have multiple inputs and multiple outputs, and these are influenced by interactions. These systems can be controlled by a centralized or decentralized controller. The main advantages of decentralization include simplifying the overall task by dividing it into a set of sub-tasks. Decentralized control is very often used in practice. Decentralized charge control of electric vehicles is a nice application. There was introduced a fully decentralized and participatory learning mechanism for privacy-preserving coordinated charging control of electric vehicles that regulates three Smart Grid socio-technical aspects: (i) reliability, (ii) discomfort and (iii) fairness [1]. Another good application of decentralized control is a quadrocopter, namely outdoor flocking of quadcopter drones with decentralized model predictive control [2]. From the theoretical point of view, there was proposed a decentralized explicit (closed-form) iterative formula that solves convex programming problems with linear equality constraints and interval bounds on the decision variables [3], or decentralized control problem for non-affine large-scale systems with nonaffine functions possibly being discontinuous [4], or decentralized adaptive tracking control strategy consisting of a steady-state controller and

modified optimal feedback controller. Design parameters-dependent feasibility conditions were formulated by using Lyapunov theory to guarantee the existence of the proposed decentralized control scheme [5]. Decentralized voltage control is another example of a decentralized strategy. It includes network partitioning strategy for the optimal voltage control of Active Distribution Networks actuated by means of a limited number of Distributed Energy Storage Systems [6]. Another paper is concerned with the problem of decentralized event-triggered dynamic output feedback control for large-scale systems with unknown interconnections. By using a modified cyclic-small gain condition and introducing a free-matrix-based integral inequality, a sufficient condition was derived to ensure that the overall closed-loop system is asymptotically stable with a prescribed H∞ performance [7]. There was also implemented fuzzy decentralized control, for example an adaptive fuzzy decentralized control approach for a class of nonlinear systems with unknown control direction and different performance constraints. In the control design, the different performance constraints, that were the prescribed performance error constraints for some subsystems and the asymmetric time-varying output constraints for the others, could be unified as one form by selecting proper performance functions [8]. Adaptive control is another area that has expanded the use of decentralized control, and here are at least a few such examples. A minimal-neural-networks-based design approach was presented for the decentralized output feedback tracking of uncertain interconnected strict-feedback nonlinear systems with unknown time varying delayed interactions unmatched in control inputs [9]. The decentralized output-feedback adaptive backstepping control scheme was also proposed for a class of interconnected nonlinear systems with unknown actuator failures by introducing a kind of high gain K-filters [10], or decentralized output-feedback adaptive control scheme for a class of interconnected nonlinear systems with input quantization. Both logarithmic quantizers and improved hysteretic quantizers were studied, and a linear time-varying model was introduced to handle the difficulty caused by quantization [11]. A decentralized adaptive backstepping control scheme was also proposed for a class of interconnected systems with nonlinear multisource disturbances and actuator faults. The nonlinear multisource disturbances comprised two parts: one was the time-varying parameterized uncertainty; the other was the dynamic unexpected signal formulated by a nonlinear exogenous system [12]. Additionally, the problem of decentralized adaptive backstepping control for a class of large-scale stochastic nonlinear time-delay systems with asymmetric saturation actuators and output constraints was also solved [13]. It can be mentioned that a backstepping-based robust decentralized adaptive neural H ∞ tracking control method was addressed for a class of large-scale strict feedback nonlinear systems with uncertain disturbances [14]. Decentralized control was implemented in robotics. The example where a discrete-time decentralized neural identification and control for large-scale uncertain nonlinear systems at a two degree of freedom planar robot was implemented can be mentioned [15], or the work that investigated the use of a decentralized control system for suppressing the vibration of a multi-link flexible robotic manipulator using embedded smart piezoelectric transducers [16]. Decentralized motion coordination algorithms were proposed for the robots so that they collectively moved in a rectangular lattice pattern from any initial position [17]. Mobile robot formations differ in accordance to the mission, environment and robot abilities. In the case of decentralized control, the ability to achieve the shapes of these formations has to be built in the controllers of each autonomous robot [18]. A decentralized control algorithm for the robots to accomplish the sweep coverage was also proposed. The sweep coverage was achieved by coordinating the robots to move along a given path that was unknown to the vehicles a priori [19].

During the simulation experiments in the real-time laboratory in recent years, we reveal the fact that some time is needed to get the appropriate behaviour of control when the self-tuning controller is used. This time depends on the type of the controlled system. This is the known problem of self-tuning control because the controller needs the adequate model of the system. This is one problem that we solve by a new approach described in

this paper, by the method named as pre-identification. Another problem comes from the fact that we used the decentralized controllers for the control of multivariable systems. If one set-point changes its value, it influences all other main subsystems by interactions and therefore the model of subsystems changes. This could be also fixed by a self-tuning controller but some time is needed to obtain the stable model. Therefore, by the new method described in this paper, named as pre-identification, we also solved this problem.

## 2. Materials and Methods

### 2.1. Decentralized Control

Using the decentralized approach, the control is divided into a set of sub-tasks that are matched by simple controllers. These partial tasks will then give us the overall course of control. The main advantages of decentralized control are primarily that a more complex system is divided into a set of simple tasks and the resulting controller is more flexible [20].

A special example of multidimensional systems is a system with two inputs and two outputs. This can be realized by the so-called P structure, see Figure 1. In this case, the inputs to the systems describing the interactions are the values of the action signals of the SISO controllers and their outputs are added to the opposite outputs of the main diagonal systems. From this figure, we get the transfer function equations of the model in the form

$$G_{S1} = \frac{Y_1}{U_1} = G_{S11} - \frac{G_{S21}G_{S12}G_{R22}}{1 + G_{S22}G_{R22}} \tag{1}$$

$$G_{S2} = \frac{Y_2}{U_2} = G_{S22} - \frac{G_{S21}G_{S12}G_{R11}}{1 + G_{S11}G_{R11}} \tag{2}$$

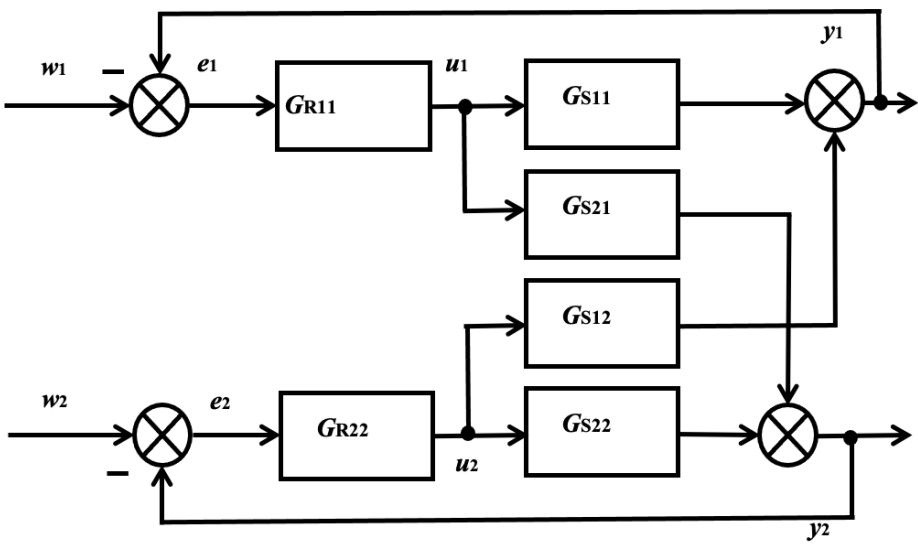

**Figure 1.** Decentralized system control with two inputs and two outputs, the so-called P structure.

### 2.2. Recursive Identification Using Approximation Polynomials

A prerequisite for good control is the most accurate description of the regulated system. Identification is the procedure by which the mathematical model of a system is obtained. The beginnings of identification based on continuous models date back to the middle of the 20th century. For continuous-time identification, the identified model is in the form of the differential equations. Differential equations contain expressions with derivatives over time that are not measurable. It is possible to replace the segment by an approximation polynomial whose derivatives can be calculated analytically in advance and then calculated numerically, see Figure 2. This approach was for example used by Perutka [20].

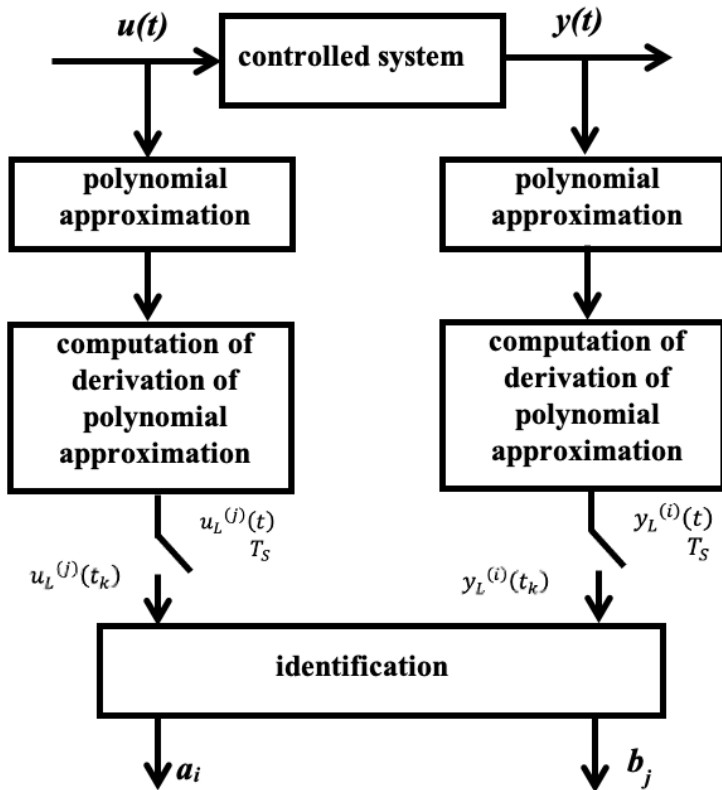

**Figure 2.** Identification scheme for continuous-time systems.

### 2.3. Least Squares Method with Exponential Forgetting

The estimation of model parameters is computed as

$$\hat{\theta}(k) = \hat{\theta}(k-1) + K(k)\hat{e}(k) \tag{3}$$

The gain vector is calculated as

$$K(k) = \frac{C(k-1)\phi(k)}{1 + \phi^T(k)C(k-1)\phi(k)} \tag{4}$$

and covariance matrix

$$C(k) = C(k-1) - \frac{C(k-1)\phi(k)\phi^T(k)C(k-1)}{1 + \phi^T(k)C(k-1)\phi(k)} \tag{5}$$

The following applies to the calculation of the prediction error

$$\hat{e}(k) = y(k) - \phi^T(k)\hat{\theta}(k-1) \tag{6}$$

In the case of exponential forgetting, the criterion of identification is

$$J = \sum_{i=k_0}^{k} \left( \varphi^{k-i} e(i) \right)^2 \tag{7}$$

where the exponential forgetting factor is chosen in the range of 0 to 1, the most common near 1.

If

$$\phi^T(k)C(k-1)\phi(k) > 0 \tag{8}$$

Then

$$C(k) = C(k-1) - \frac{C(k-1)\phi(k)\phi^T(k)C(k-1)}{\eta^{-1} + \phi^T(k)C(k-1)\phi(k)} \tag{9}$$

where

$$\eta(k) = \varphi(k) - \frac{1 - \varphi(k)}{\xi(k)} \tag{10}$$

If

$$\phi^T(k)C(k-1)\phi(k) = 0 \tag{11}$$

Then

$$C(k) = C(k-1) \tag{12}$$

Furthermore

$$\varphi(k) = \left\{ 1 + (1+\rho)[\ln(1+\xi(k-1))] + \left[ \frac{(v(k-1)+1)\eta(k-1)}{1+\xi(k-1)+\eta(k-1)} - 1 \right] \frac{\xi(k-1)}{1+\xi(k-1)} \right\}^{-1} \tag{13}$$

$$\eta(k) = \frac{e^2(k)}{\lambda(k)} \tag{14}$$

$$v(k) = \varphi(k)[v(k-1) + 1] \tag{15}$$

$$\lambda(k) = \varphi(k)\left[ \lambda(k-1) + \frac{e^2(k)}{1+\xi(k-1)} \right] \tag{16}$$

$$\xi(k) = \phi^T(k)C(k-1)\phi(k) \tag{17}$$

The parameters estimation vector is in the form

$$\hat{\theta}^T(k) = \left( \hat{a}_0, \hat{a}_1, \ldots, \hat{a}_{\deg(a)}, \hat{b}_0, \hat{b}_1, \ldots, \hat{b}_{\deg(b)}, d \right) \tag{18}$$

and regressor

$$\phi^T(k) = \left( -y(t_k), \ldots, -y_L^{(n-1)}(t_k), u(t_k), \ldots, u_L^{(m)}(t_k), 1 \right) \tag{19}$$

### 2.4. Self-Tuning Controller

The main reason for using adaptive control is that the systems change over time or the characteristics of the controlled system are unknown. The basic principle of adaptive systems is to change the characteristics of the controller based on the characteristics of the controlled process [21]. The general scheme of the self-tuning controller is shown in Figure 3.

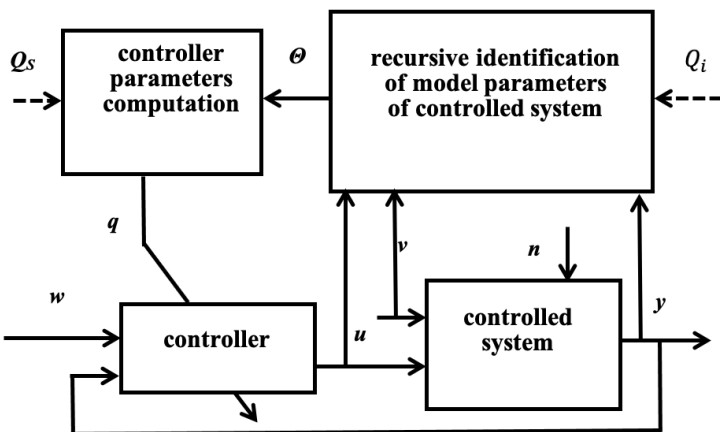

**Figure 3.** The general scheme of the self-tuning controller.

### 2.5. Suboptimal Linear Quadratic Tracking Controller

The method was introduced by Dostál [22]. If the system of Figure 4 is considered

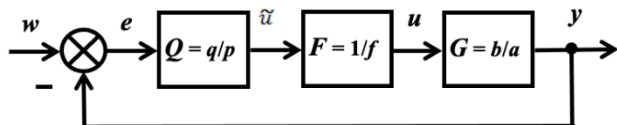

**Figure 4.** System with feedback controller.

Let us minimize a quadratic functional with two penalty constants

$$J = \int_0^\infty \left\{ \mu e^2(t) + \varphi \tilde{u}^2(t) \right\} dt \tag{20}$$

The Laplace image of the set point holds

$$w(s) = \frac{h_w(s)}{s f_w(s)} \tag{21}$$

It holds for degrees of polynomials

$$\deg(h_w) \leq \deg(f_w), \; f_w(0) \neq 0 \tag{22}$$

We calculate stable polynomials g and $n$ as results of spectral factorizations

$$(as) * \varphi as + b * \mu b = g * g, n * n = a * a \tag{23}$$

We solve the following diophantine equation

$$asp + bq = gn \tag{24}$$

Considering the transfer function of the system

$$G(s) = \frac{b_0}{s^2 + a_1 s + a_0} \tag{25}$$

then the controller is

$$F(s)Q(s) = \frac{q_2 s^2 + q_1 s + q_0}{s(p_2 s^2 + p_1 s + p_0)} \tag{26}$$

In this case, the polynomials have the form

$$g(s) = g_3 s^3 + g_2 s^2 + g_1 s + g_0 \tag{27}$$

$$n(s) = s^2 + n_1 s + n_0 \tag{28}$$

and to calculate their coefficients obtained by spectral factorization

$$g_0 = \sqrt{\mu b_0^2} \tag{29}$$

$$g_1 = \sqrt{2 g_2 g_0 + \varphi a_0^2} \tag{30}$$

$$g_2 = \sqrt{2 g_3 g_1 + \varphi (a_1^2 - 2 a_0)} \tag{31}$$

$$g_3 = \sqrt{\varphi} \tag{32}$$

$$n_0 = \sqrt{a_2^0} \tag{33}$$

$$n_1 = \sqrt{2n_0 - a_1^2 - 2a_0} \tag{34}$$

### 2.6. Calculation of Derivatives Using Approximation Functions

To calculate the derivatives, we approximate the closest neighborhood for a given time by the approximation function. For example, we will use the Lagrange polynomial in the form

$$P_2(x) = \frac{(x-b)(x-c)}{(a-b)(a-c)}f(a) + \frac{(x-a)(x-c)}{(b-a)(b-c)}f(b) + \frac{(x-b)(x-b)}{(c-a)(c-b)}f(ac) \tag{35}$$

whose first derivative is

$$f'(x) \cong P_2'(x) = \frac{2x-(b+c)}{(a-b)(a-c)}f(a) + \frac{2x-(a+c)}{(b-a)(b-c)}f(b) \\ + \frac{2x-(a+b)}{(c-a)(c-b)}f(c) \tag{36}$$

and second derivative is

$$f''(x) \cong P_2''(x) = \frac{2f(a)}{(a-b)(a-c)} + \frac{2f(b)}{(b-a)(b-c)} + \frac{2f(c)}{(c-a)(c-b)} \tag{37}$$

### 2.7. Recursive Instrumental Variable Method

The instrumental variable method is a modification of the least squares method. The least squares method uses the quadratic criterion and the existence of one global minimum. However, a prerequisite for successful least-squares modelling is that the fault is represented by white noise with zero mean value. The instrumental variable method does not make it possible to determine the noise properties, but is based on weaker assumptions than the least squares method. The identification proceeds according to number 5. As with the least squares method, the method of the instrumental variable can also be formulated recursively [22–25]. The parameter estimation vector has the form

$$\hat{\theta}^T(k) = \left(\hat{a}_0, \hat{a}_1, \ldots, \hat{a}_{\deg(a)}, \hat{b}_0, \hat{b}_1, \ldots, \hat{b}_{\deg(b)}, d\right) \tag{38}$$

and data vector

$$\hat{\theta}^T(k) = (-y(t_k), \ldots - y_L^{(n-1)}(t_k), u(t_k), \ldots, u_L^{(m)}(t_k), 1 \tag{39}$$

The gain vector is calculated by relation

$$L(k) = \frac{C(k-1)z(k)}{1 + \phi^T(k)C(k-1)z(k-1)} \tag{40}$$

In addition to the data vector, it is necessary to know the covariance matrix

$$C(k) = C(k-1) - \frac{C(k-1)z(k)\phi^T(k)C(k-1)}{1 + \phi^T(k)C(k-1)z(k)} \tag{41}$$

and instrument vector

$$z(k) = (u(t_k), u(t_{k-1}), \ldots, u(t_{k-n-m}), \tag{42}$$

which we choose as a set of delayed inputs. The prediction error is calculated by

$$\hat{e}(k) = y(k) - \phi^T(k)\hat{\theta}(k-1) \tag{43}$$

and estimating the parameters according to

$$\hat{\theta}(k) = \hat{\theta}(k-1) + L(k)\hat{e}(k) \tag{44}$$

### 3. Assumptions

*3.1. Decentralized Controllability*

Assume the existence of a stable minimum phase Linear Time Invariant (LTI) in time of a continuous multidimensional system of the dimension $N \times N$. Its Laplace transformation $S(s) : S(t) > S(s)$, which we call the transformed system is in the form

$$S(s) = \begin{pmatrix} S_{11}(s)S_{12}(s) & \cdots & S_{1N}(s) \\ \vdots & \ddots & \vdots \\ S_{N1}(s)S_{N2}(s) & \cdots & S_{NN}(s) \end{pmatrix} \tag{45}$$

where $S_{ij}(s)$, $i = 1, 2, \ldots, N$, $j = 1, 2, \ldots$ is Laplace transformation of the *ij*-th subset of $S_{ij}(t)$ of the transformed system $S(s)$. The transformed system $S(s)$ je is decentrally controllable only when its main diagonal is dominant.

*3.2. System Model and Shape of Reference Signal*

Suppose there exists a system $S(t)$ and a transformed system $S(s)$ as described above. Then we formulate a model created for the purpose of decentralized control, which we call $M(t)$, and its Laplace transformation $M(s)$. Consider $M(s)$ as a stable minimum phase linear time invariant multivariate diagonal matrix in the form

$$M(s) = \begin{pmatrix} M(s) & 0 & \cdots & 0 \\ 0 & M(s) & \cdots & 0 \\ \vdots & \vdots & \ddots & \vdots \\ 0 & 0 & \cdots & M_{N(s)} \end{pmatrix} \tag{46}$$

where $M_i(s)$, $i = 1, 2, \ldots, N$ is the Laplace transformation of the *i*-th submodule $M_i(t)$ of the model $M(t)$ of the transformed system $S(s)$. This assumes minimal impact of extra-diagonal transmissions, which is important because of the deployment of a decentralized controller. Simplification of the *N*-dimensional system to *N*-dimensional systems is simplified.

Furthermore, suppose that the Laplace transformation of the reference signal vector $r(s)$ is always in the form

$$r(s) = (R_1(s), R_2(s), \ldots, R_N(s)) \tag{47}$$

where $R_i(s)$, $i = 1, 2\ldots, N$ is the *i*-th Laplace reference signal of Laplace transformation of the reference signal vector $r(s)$ and has the form

$$R_i(s) = \frac{h_i}{s} \tag{48}$$

where $h_i \in R$, $i = 1, 2, \ldots, N$, is the *i*-th part by constant function, i.e., reference signal, which is only a combination of *p-l* step changes of the signal of its different constant values

$$h_i = \begin{cases} j_{i1} \; for \; t \; \in \; \langle 0, \; t_{i1} \rangle \\ j_{i2} \; for \; t \; \in \; \langle t_{i1}, \; t_{i2} \rangle \\ \qquad \vdots \\ j_{ip} \; for \; t \; \in \; \langle t_{ip-1}, \; t_{ip} \rangle \end{cases}, \; 0 < t_{i1} < t_{i2} < \cdots < t_{ip-1} < t_{ip} \tag{49}$$

where $j_{il} \in R$, $i = 1, 2, \ldots, N$, $l = 1, 2, \ldots, p$, the *l*-th constant function *i*-th in parts by the constant function $h_i$, $t$ is the time $t_{il} \in R^+$, $i = 1, 2, \ldots, N$, $l = 1, 2, \ldots, p$, the *l*-th moment of the *i*-th in portions of the constant function $h_i$. This means that each non-zero element of the matrix $M(s)$ has exactly one non-zero element of the vector $r(s)$, i.e., that each partial transmission of the overall system model has a reference signal defined for it. As for the form of the reference signal, it is a constant function in parts. This function

is approximated from an arbitrary but predetermined number of $p$ segments of a different but concise value, i.e., it varies over time.

## 4. Pre-Identification

Consider the validity of the assumption of decentralized controllability, system description and system model. Then, the transformed system S($s$) can be viewed as a black box model, and let it be identified by direct and/or indirect time-continuous algorithms. In time, continuous model identification can be realized by following steps: The controller is not connected in the closed circuit. The values of the vector of difference of output quantities and reference signals E($t$) are sent to the input of the system S($t$). The values of the reference signals are the same and at the same time as those that will be used during regulation.

1. The controller is not connected in the closed circuit. The values of the vector of difference of output quantities and reference signals E($t$) are sent to the input of the system S($t$). The values of the reference signals are the same and at the same time as those that will be used during regulation.
2. If switching control is considered, each time interval of the control of the system S($s$) at which all reference signals have a constant value is identified separately, in so-called Identification Elements (IE).
3. Each identification element is identified several times, each time by a different identification algorithm, and the obtained model can be verified by comparison with the measured data. The obtained model, which is most consistent with the measured data, is then used for control. Let us call this method of Identification More Than One Method (IMTOM).

The system S($s$) is completely identified by the above procedure before the actual regulation begins, therefore identification during the regulation is not necessary. This procedure is suitable for processes where the same procedure is repeated many times.

## 5. Results

### 5.1. Simulation Results

The verified system is described as

$$G_S(s) = \begin{pmatrix} \frac{3.7}{s^2+5.2s+4.6} & \frac{0.4}{s^2+4.4s+3.8} \\ \frac{0.6}{s^2+10.6s+10.2} & \frac{8.7}{s^2+7.4s+8.2} \end{pmatrix} \tag{50}$$

Since it is the system with two inputs and two outputs, we obtained two suboptimal linear quadratic controllers in the form that was described in the previous section. These controllers had the following penalty constants

$$\mu_1 = 1, \ \varphi_1 = 30, \mu_2 = 1, \ \varphi_2 = 30 \tag{51}$$

We used our algorithm, pre-identification, at this system and we obtained the following results, see Figures 5–8. First, we performed system response on the given set-points depicted in Figure 5. Together with this system response, we obtained the system pre-identified parameters shown in Figure 6, for the first subsystem in the left one and for the second subsystem in the right one. According to these pre-identified parameters we performed the simulation of control which is shown in Figure 7. During the control, the controller parameters were changing, and they are recorded in Figure 8, for the first controller in left one and and for the second controller in the right one.

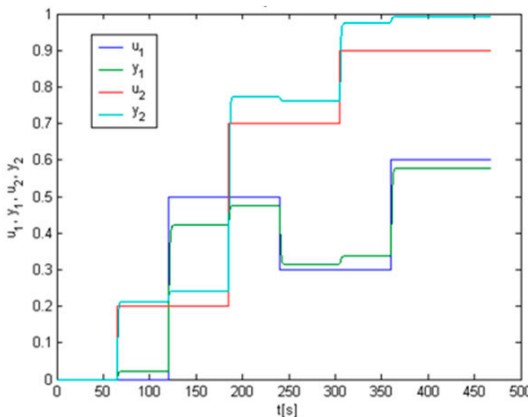

**Figure 5.** System response (green and cyan) on the given set-points (blue and red).

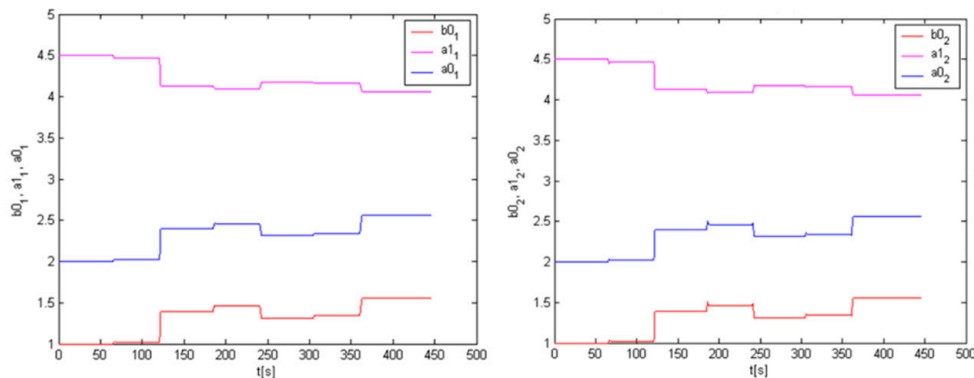

**Figure 6.** System pre-identification using recursive intrumental variable—1st subsystem (**left**), 2nd (**right**).

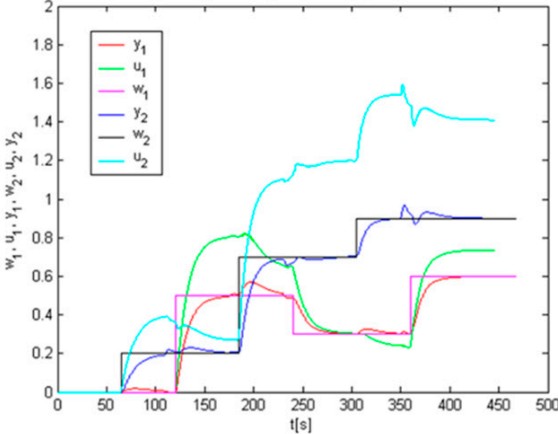

**Figure 7.** Output of control.

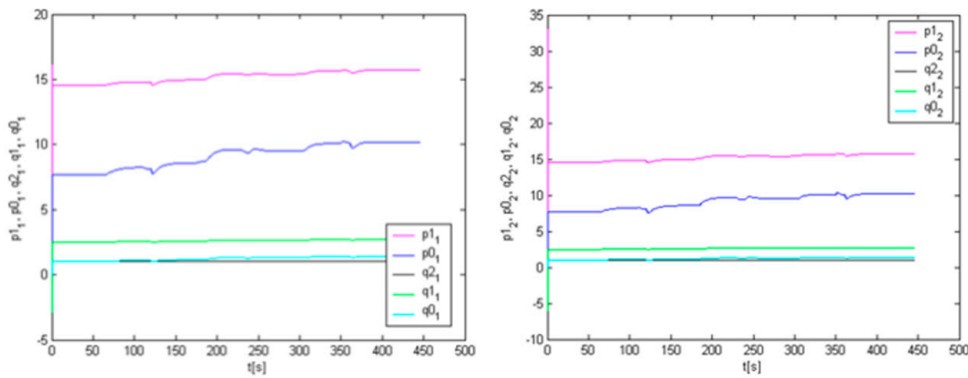

**Figure 8.** Controllers parameters during simulation—1st (**left**), 2nd (**right**).

### 5.2. Results in Real-Time at Laboratory Model

We verified the presented method in real-time control using MATLAB at CE108 Coupled Drives Apparatus Model [26] which is shown in the Figure 9. The laboratory model CE108 allows solving practical problems of tensioning and speed of material movement in continuous processes. An example is the speed and tension of the thread when rewinding from one spool to another, which must be controlled. This situation is adapted for laboratory measurements so that the elastic band is mounted on three wheels. The lower two wheels are fixed, their speed can be measured and regulated. The third wheel can move (located on a movable arm suspended on a spring) and simulates a workstation along with tension and speed measurements. Two servomotors control the speed of movement and tensioning of the belt. The speed is 0–3000 rpm, which corresponds to a voltage of 0–10 V. Tension measurement is indirect through the angle of the movable arm, from $-10°$ to $10°$, which corresponds to a voltage from $-10$ V to $10$ V. Inputs and outputs are located on the front panel of the device; it is the control voltage to the servomotor amplifiers, which are bidirectional, and which are inputs. There are four outputs, the voltage corresponding to the speed of the two servomotors and the voltage corresponding to the tension and the speed of the belt.

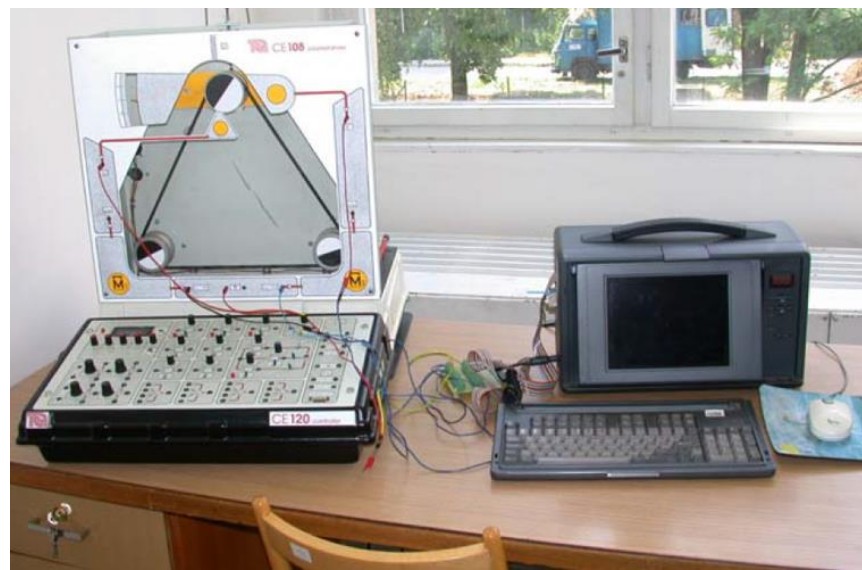

**Figure 9.** Photo of CE108 Coupled Drives Apparatus model connected with PC with MATLAB.

Using the pre-identification method and fully implementing interactions in the main plants, we obtained the following mathematical model to be used at control of speed

$$G_S(s) = \begin{pmatrix} \frac{0.78}{s^2+2.66s+1.33} & 0 \\ 0 & \frac{4.16}{s^2+11.66s+16.66} \end{pmatrix} \tag{52}$$

Since it is the system with two inputs and two outputs, we obtained two suboptimal linear quadratic controllers in the form that was described in the previous section. These controllers had the following penalty constants

$$\mu_1 = 1, \; \varphi_1 = 1, \mu_2 = 1, \; \varphi_2 = 0.85 \tag{53}$$

We used our algorithm, pre-identification, at this system and we obtained the following results, see Figures 10–13. First, we performed system response on the given set-points depicted in Figure 10. Together with this system response, we obtained the system pre-identified parameters shown in Figure 11, for the first subsystem in the left one and for the second subsystem in the right one. According to these pre-identified parameters we performed the simulation of control which is shown in Figure 12. During the control, the controller parameters were changing, and they are recorded in Figure 13, for the first controller in left one and and for the second controller in the right one.

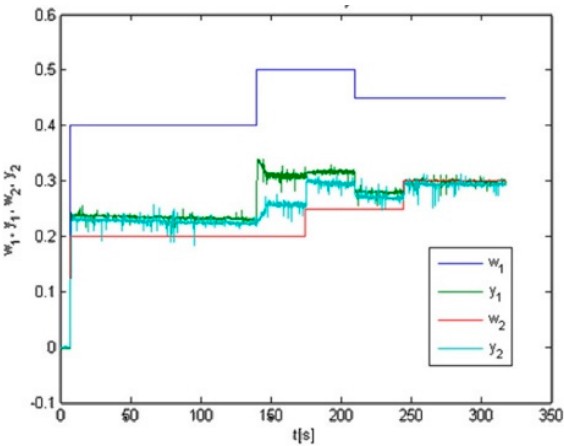

**Figure 10.** System response (green and cyan) on the given set-points (blue and red).

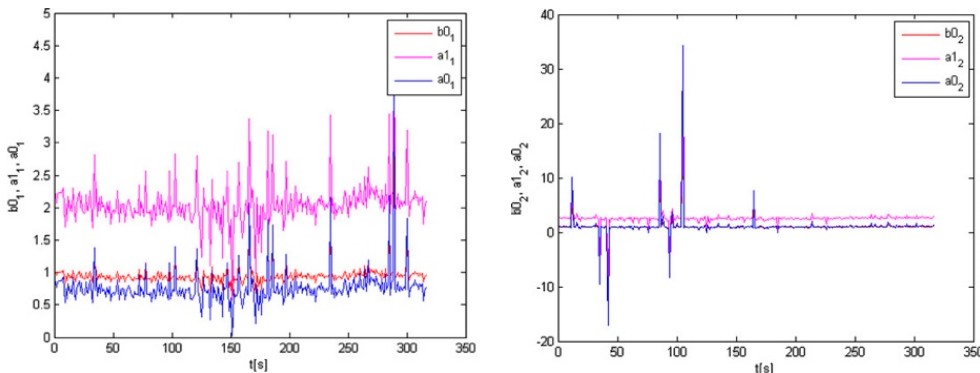

**Figure 11.** System pre-identification using recursive instrumental variable—1st subsystem (**left**), 2nd (**right**).

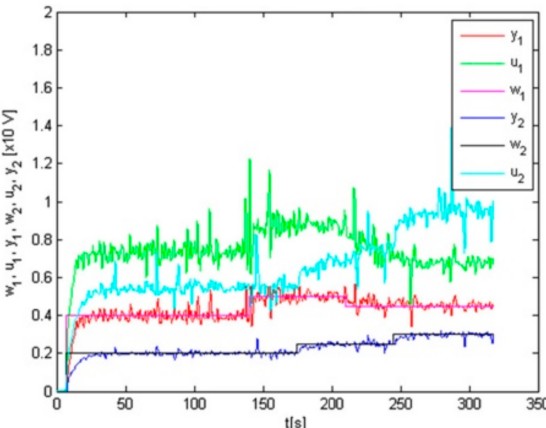

**Figure 12.** Output of control.

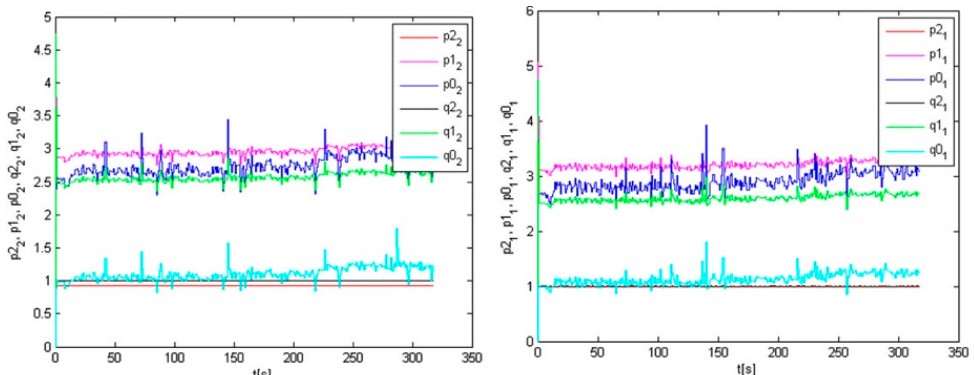

**Figure 13.** Controllers parameters during simulation—1st (**left**), 2nd (**right**).

## 6. Conclusions

This paper presents the new method of identification named as pre-identification on the theoretical level and subsequently verified it by simulations and in the real-time experiments at Coupled Drives model in the laboratory. The results confirm that the method can be successfully used and future work will focus on the verification of this method on more examples both in simulation and in laboratory conditions. This new method enhances the usage of a decentralized self-tuning controller in the way that it fixes the time the adaptive controller needs to adapt its model.

**Author Contributions:** Conceptualization, K.P. and I.K.; methodology, K.P. and I.K.; software, I.Z.; validation, V.B., K.P. and V.T.; formal analysis, M.B.; investigation, K.P. and V.T.; resources, K.P.; data curation, K.P. and M.B.; writing—original draft preparation, K.P.; writing—review and editing, I.Z.; visualization, K.P. and I.Z.; supervision, K.P. and M.S.; project administration, K.P. and I.K.; funding acquisition, M.S. All authors have read and agreed to the published version of the manuscript.

**Funding:** This research was funded by by the European Regional Development Fund under the project CEBIA-Tech No. CZ.1.05/2.1.00/03.0089. This research was funded by the Ministry of Education, Science, research and Sport of the Slovak Republic under the project STIMULY MATADOR 1247/2018. Project title: Research and development of modular reconfigurable production systems using Smart Industry principles for automotive with pilot application in MoBearing Line industry.

**Institutional Review Board Statement:** Not applicable.

**Informed Consent Statement:** Not applicable.

**Data Availability Statement:** Not applicable.

**Acknowledgments:** This work was supported by the European Regional Development Fund under the project CEBIA-Tech No. CZ.1.05/2.1.00/03.0089. This article was made under support of project: STIMULY MATADOR 1247/2018. Project title: Research and development of modular reconfigurable production systems using Smart Industry principles for automotive with pilot application in MoBearing Line industry.

**Conflicts of Interest:** The authors declare no conflict of interest.

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
