# Peer review of "Methods of Pre-Identification of TITO Systems"

_applsci, doi:10.3390/app11156954_

Round 1

Reviewer 1 Report

The paper presents a solution for system automation for identifications. The problem statement and the goal of the paper is not defiend. In lack of goal, the paper has no significant contribution and shall be rejected. 

The Introduction gives a long and detailed description of centralized and decentralized control system. On the other hand, there is no problem stated nor the goal of the research determined. The methods part gives a conceptional overview of decentralized control systems. 

The results presents calculatiosn and simulations in a selected system. The results presents only a single figure about the results and vaguely shows that there were multiple simulations run. 

The abstract is too short and do not presents the contribution of the paper. 

To sum up, the paper in this current form shall be rejected. 

Author Response

Thank you very much for your valuable comments and recommendations on our paper. I edited the paper in accordance with your recommendations.

Reviewer 2 Report

The article describes the established methods of identification of systems with two inputs, two outputs, and with two interactions. In the introduction, the authors analyze in detail the possibilities and use of these systems in practice.
The simulation results are described too briefly. I propose to expand them and describe them in more detail. This is especially Chapter 5 - Results. It would also be appropriate to focus on the practical idea of ​​using the results.

Specific suggestions for repairs:
1 / Remove the lines around all formulas.
2 / Figure 2 - Remove the red underlined wavy lines under the variables ui and yi.
3 / Figure 3 - Move the text between Figure 3. and the image itself.
4 / Figure 3 - Remove the red mark at the top right of Qi.
5 / I recommend defining the abbreviation TITO already in the introduction, or consider the definition in the abstract.
6 / In chapter 3.1, increase the first letters of the words LTI. Similar to e.g. Two-Input Two-Output (TITO).
7 / Line 278 - Increase the initial letters EI.
8 / Line 282 - Increase the initial letters IVM.
9 / Line 267 - Point 1 should not be from "The controller ...."?
10 / Figure 5 - it is necessary to enlarge the descriptions of the axes. This is how they are illegible.

Author Response

(The authors gave the same response as above.)

Reviewer 3 Report

The authors have presented the control systems on a theoretical level and was explained in-depth. However,a real time testing is required in order to validate the theory. Also, the introduction could have been reduced as some information seems too detailed which could have been removed.

Author Response

(The authors gave the same response as above.)
